**Subject Category:**
Biology (whole organism)

behaviour/ecology/evolution

animal culture, cultural evolution, song, cetacean, humpback whale, south pacific

**Author for correspondence:**
Ellen C. Garland
e-mail: ecg5@st-andrews.ac.uk

# Migratory convergence facilitates cultural transmission of humpback whale song

Clare Owen[1], Luke Rendell[1,2], Rochelle Constantine[3,4], Michael J. Noad[3,5], Jenny Allen[5], Olive Andrews[3,6,7], Claire Garrigue[3,8,9], M. Michael Poole[3,10], David Donnelly[3,11], Nan Hauser[3,12] and Ellen C. Garland[1,2,3]

[1]Sea Mammal Research Unit, School of Biology, University of St Andrews, St Andrews KY16 8LB, UK
[2]Centre for Social Learning and Cognitive Evolution, School of Biology, University of St Andrews, St Andrews KY16 9TH, UK
[3]South Pacific Whale Research Consortium, PO Box 3069, Avarua, Rarotonga, Cook Islands
[4]School of Biological Sciences, Institute of Marine Science, University of Auckland, Private Bag 92019, Auckland 1142, New Zealand
[5]Cetacean Ecology and Acoustics Laboratory, School of Veterinary Science, The University of Queensland, Gatton, Queensland 4343, Australia
[6]Conservation International, Pacific Islands Programme, Science Building 302, University of Auckland, 23 Symonds Street, Auckland 1010, New Zealand
[7]Niue Whale Research Project, Alofi, Niue
[8]Opération Cétacés, Noumea 98802, New Caledonia
[9]UMR ENTROPIE (IRD, Université de La Réunion, CNRS, Laboratoire d'excellence-CORAIL), BPA5, 98848 Noumea Cedex, New Caledonia
[10]Marine Mammal Research Program, BP 698, Maharepa, 98728 Moorea, French Polynesia
[11]Killer Whales Australia, 8 Campbell Parade, Box Hill South, Victoria 3128, Australia
[12]Cook Islands Whale Research, PO Box 3069, Avarua, Rarotonga, Cook Islands

LR, 0000-0002-1121-9142; JA, 0000-0003-4658-7380; ECG, 0000-0002-8240-1267

Cultural transmission of behaviour is important in a wide variety of vertebrate taxa from birds to humans. Vocal traditions and vocal learning provide a strong foundation for studying culture and its transmission in both humans and cetaceans. Male humpback whales (*Megaptera novaeangliae*) perform complex, culturally transmitted song displays that can change both evolutionarily (through accumulations of small changes) or revolutionarily (where a population rapidly adopts a novel song). The degree of coordination and conformity underlying song revolutions makes their study of particular interest. Acoustic contact on migratory routes may

provide a mechanism for cultural revolutions of song, yet these areas of contact remain uncertain. Here, we compared songs recorded from the Kermadec Islands, a recently discovered migratory stopover, to multiple South Pacific wintering grounds. Similarities in song themes from the Kermadec Islands and multiple wintering locations (from New Caledonia across to the Cook Islands) suggest a location allowing cultural transmission of song eastward across the South Pacific, active song learning (hybrid songs) and the potential for cultural convergence after acoustic isolation at the wintering grounds. As with the correlations in humans between genes, communication and migration, the migration patterns of humpback whales are written into their songs.

# 1. Introduction

Cultural traditions play a significant role in shaping human societies [1]. Culture, broadly defined as shared behaviour or information within a community acquired through some form of social learning from conspecifics [2,3], can also be an important process in non-human communities, as demonstrated in multiple studies on the cultural learning of behavioural traits in primates [4]. Recent studies have also highlighted the importance of cultural traits and social learning in cetaceans [3,5,6]. For example, bottlenose dolphins (*Tursiops* sp.) demonstrate the cultural transmission of tool use [7], while humpback whales (*Megaptera novaeangliae*) have multiple, independently evolving cultural traditions including migratory destinations, feeding techniques and songs [8–11]. Vocal traditions and vocal learning provide a strong foundation for studying culture and its transmission in both humans and cetaceans [5]. For example, vocalizations that are shared within a group and maintained through cultural transmission over decades are used to identify social structures in killer whales (*Orcinus orca*) [12] and sperm whales (*Physeter macrocephalus*) [13]. The stability of these vocal cultures and other cultural traditions (e.g. prey specialization) can in turn affect genetic evolution through gene-culture coevolutionary processes [3].

A striking example of large-scale cultural transmission in a non-human animal is the transmission of humpback whale song between populations [9]. Male humpback whales produce long, stereotyped vocalizations [14] that function in sexual selection for mate attraction and/or to facilitate male–male interactions [15]. Humpback whale song is hierarchically structured: sound units are grouped into phrases that are embedded in higher-level themes [14]. Although songs are constantly evolving, most males within a population will converge on a single song type during any particular winter breeding season [16,17]. Songs can also be transmitted between populations. Garland *et al.* [9] identified dynamic transmission of humpback whale song that extended across the South Pacific, spanning 6000 km from eastern Australia in the west to French Polynesia in the east. It is a clear example of large-scale horizontal cultural transmission, where a population rapidly adopts a novel song introduced from a neighbouring population [11], and then the next adjacent population adopts the novel song, and so on in a population-level transmission chain [9].

The western and central South Pacific region can be divided into three sections: eastern Australia, western South Pacific (New Caledonia, Tonga and American Samoa) and the central South Pacific (Cook Islands and French Polynesia) based on previous song studies [18]. Song types take approximately 2 years to transit across the region and as a result, populations in each section will typically converge on different song types at any one time; these songs may be related to each other with some shared material or be quite distinct [9,11,19,20]. However, given humpback whales' fidelity to natal wintering grounds [10], we still have a limited understanding of the underlying mechanism(s) driving this cultural phenomenon.

Humpback whale song is most frequently produced and recorded on the winter breeding grounds [15] and while the whales are migrating to and (particularly in the South Pacific) from their wintering grounds [11]. Aggregations (e.g. on feeding or wintering grounds) and shared migratory routes may provide an opportunity for acoustic contact, which is necessary for song transmission. Four possible contact mechanisms have been suggested: singing on shared feeding grounds, singing on shared or partially shared migratory routes, between-season movement of individuals between populations and within-season movement of individuals between populations [16]. All these mechanisms could occur in the South Pacific [19], but capturing such events and/or identifying important geographical locations in Oceania (western and central South Pacific) remains challenging given the open ocean migratory range of humpback whales.

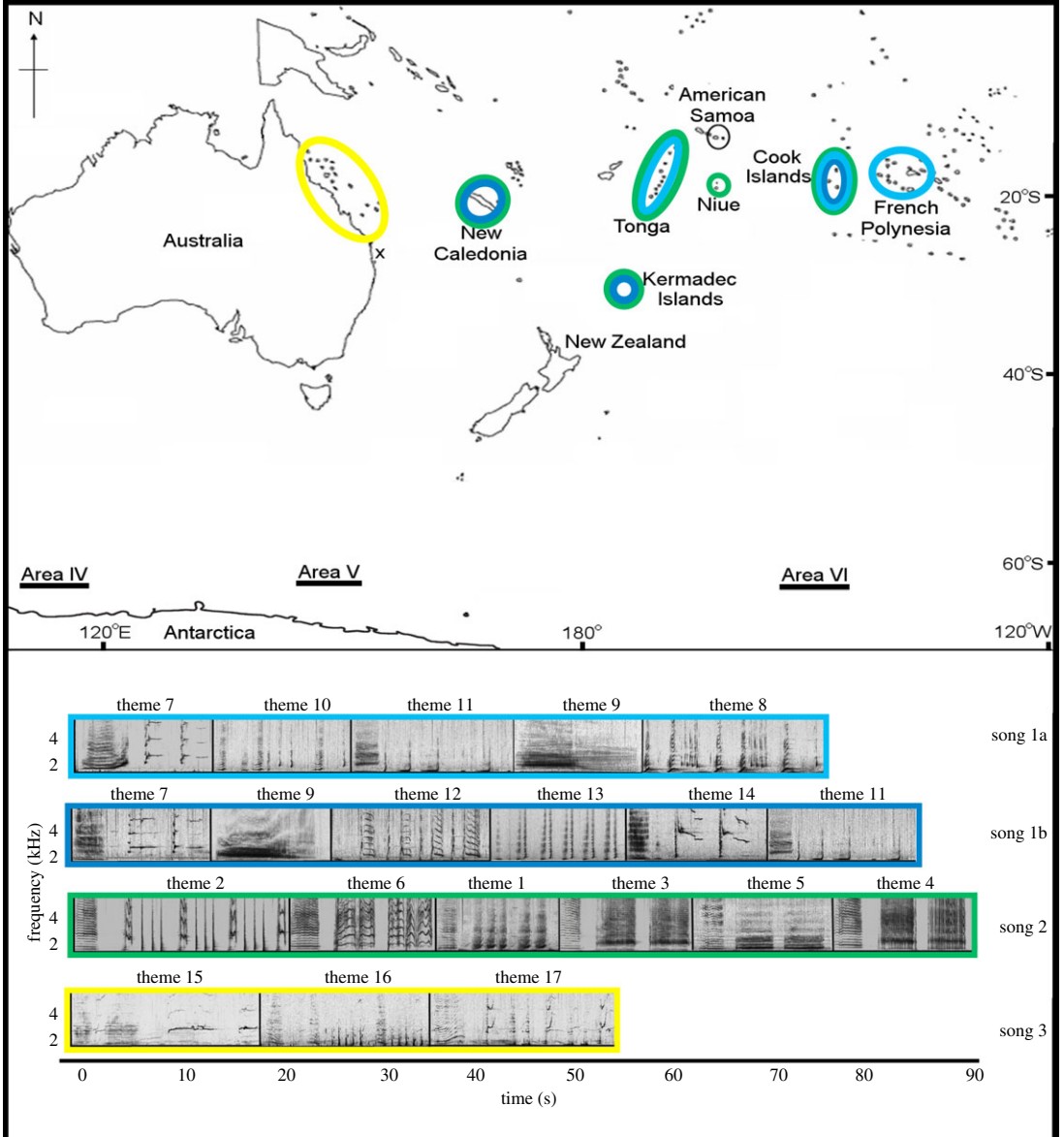

**Figure 1.** Map of the South Pacific with the wintering grounds, with the Kermadec Islands migratory stopover and the Antarctic summer feeding areas noted. The spectrograms of song types 1a and 1b (themes 7–14), song type 2 (themes 1–6) and song type 3 (themes 15–17) are colour coded to show where each song type was present. The distance between eastern Australia and French Polynesia is approximately 6000 km. *x* denotes the location of the eastern Australian recordings. Map modified from Garland *et al.* [19]. A representative phrase for each theme is shown in the typical order they were sang. Note that some themes contained variant phrases (termed 'phrase types'); a single example from the theme is shown (see electronic supplementary material, table S3 for detailed description of all phrase types). Spectrograms were 2048-point [1024-point song type 3] fast Fourier transform (FFT), Hann window, 75% overlap, displaying 5 kHz and 90 s, generated in Adobe Audition. Corresponding audio files are provided for each song type (electronic supplementary material, audio S1–S4).

A recent study at the Kermadec Islands, a remote group of islands located in the western South Pacific (figure 1), found a large number of whales were present during the Austral spring [21]. Genetic and photo-identification matches reveal the whales originated from multiple South Pacific wintering grounds [21]. With very few island groups south of the tropical wintering grounds, this unusual migratory stopover, where multiple migratory corridors may overlap when the whales are almost 2000 km into their southward migration, may provide a location for acoustic contact among multiple populations. Whales stay on average 4.6 days at Raoul Island, Kermadecs (R. C. unpublished data, 2015), which potentially allows for the exposure to song for several days. Such contact could facilitate the cultural convergence of song in the western South Pacific populations, and the easterly transmission of song into the central South Pacific.

Here, we hypothesized that if males do migrate past the Kermadec Islands from multiple wintering grounds during their southward migration (September and October 2015), we should see some evidence of the cultural processes, song transmission and/or convergence. This would provide evidence for a cultural mechanism allowing easterly transmission of song from the western to the central South Pacific region. We compared recordings from the Kermadecs with song recordings collected during the same winter breeding months (July–October 2015) from six wintering grounds spread across the South Pacific (eastern Australia, New Caledonia, Tonga, Niue, the Cook Islands and French Polynesia) and assigned a possible origin population for each Kermadec song recording (or parts thereof).

# 2. Material and methods

## 2.1. Song recordings

Humpback whale song recordings were collected near Raoul Island, Kermadecs (figure 1) in September and October 2015 during a dedicated humpback whale research voyage. A hydrophone (HTI 96MIN, frequency response 2 Hz to 30 kHz) suspended to approximately 10 m deep in water depths of approximately 60 m from a 5 m RIB was used to collect acoustic recordings when humpback whales were observed. The recordings were made on a Zoom or a Microtrack II digital recorder (WAV format, 16 bit, sampling rate 44.1 kHz). A total of 12 recordings were collected over 13 days, all within 5 km of Raoul Island. To maximize data quality and reduce uncertainty, we excluded all low-quality sections of recordings where songs were unclear (electronic supplementary material, tables S1 and S2). This resulted in a disparity between the number of recordings and the total number of 'individual' singers, as we were conservative in linking sections of song together within a recording (see below).

Songs were recorded at each wintering ground (eastern Australia: Peregian Beach, Queensland; New Caledonia: Southern Lagoon; Tonga: Tongatapu; Niue: Main Island; Cook Islands: Rarotonga; French Polynesia: Mo'orea) across the western and central South Pacific in the 2015 austral winter or spring (electronic supplementary material, table S1 and figure 1). Humpback whale song recordings from eastern Australia were collected using a moored, T-shaped, five buoy hydrophone array in 18–28 m water depth (16 bit, WAV files, 22 kHz sampling rate) as part of concurrent field research (see Noad et al. [22] and Dunlop et al. [23] for further details). Such an arrangement allowed accurate tracking of singing whales within a 10 km radius [22]. Some recordings in New Caledonia were collected using a single, moored passive acoustic recorder (SM2M + Wildlife Acoustics) in approximately 60 m water depth (22 kHz sampling rate) as part of other research. One recording from Niue was taken from a video collected using an underwater camera (Canon EOS 10D Digital SLR, audio extracted at a sampling rate of 44.1 kHz, 16 bit, WAV file format). All other recordings were made using a hand-held hydrophone (HighTech HTI 96MIN) suspended to approximately 10 m deep in water depths of 30 m to greater than 1 km connected to a digital recorder (M-Audio Microtrack 24/96 or Microtrack II, or a Zoom H2 or H4, recording at 44.1 kHz, 16 bit, WAV file format) from a designated research vessel or a platform of opportunity. While we endeavoured to analyse all available recordings to capture as much singer variability as possible, the available data represent a snapshot in time of a small (but broadly representative, due to strong song matching [9]) sample of singers from each location.

## 2.2. Song transcription

Song is hierarchical: a sequence of sounds termed 'units' comprises a 'phrase', phrases are repeated to form a 'theme' and a few themes are sung in a set order to form a 'song' [14]. At the upper level, different versions of the song that contain different themes are called 'song types' [9]. All humpback whale song recordings were analysed as spectrograms created in Adobe Audition (Fast Fourier Transform [FFT] 2048 [eastern Australia: 1024], Hann window, 75% overlap, displaying 30 s and 0–5 kHz). The song was transcribed by a human classifier (C.O.) at the unit level (following Garland et al. [9,20,24]). Individual units were named and identified by their visual and aural characteristics. Every unit was transcribed and presented as a string of units for each phrase. Each stereotyped string of units that were identified as a phrase was allocated a letter (e.g. A, B, C). Phrases containing similar sounds in the same order were grouped into the same theme number (e.g. 1, 2, 3). The qualitative assignment of phrase types (and higher-level theme groupings) was validated through clustering of unit sequences (see below). When multiple singers were present in a recording or a

singer was silent for more than 3 min before the song continued, it was not always possible to confirm the same singer was resuming. To avoid ambiguity, the subsequent phrases were labelled with a letter (e.g. KI01S1a) and the strings of phrases were analysed separately. This resulted in 39 'individual' singers in the Kermadec dataset from the 12 initial recordings (electronic supplementary material, table S1).

## 2.3. Quantifying units

To reduce subjectivity and aid in a robust, repeatable unit and theme classification, the highest quality recording at each recording location was selected to measure units for subsequent inclusion in a quantitative analysis, aiming to maximize consistent naming/classification of units between locations and song types. All units in the first phrase of each theme were measured for 11 frequency and duration measurements following previously published studies on humpback whale unit classification [23–25] using Raven Pro 1.4 (Cornell Lab of Ornithology; smoothed spectrogram, 11 s long, 21.5 Hz resolution, FFT 2048 [eastern Australia: 1024], Hann window, 75% overlap). If a unit type was not present in these selected phrases (i.e. rare units), it was located and measured to ensure all unit types identified at each location had a corresponding set of measurements. A selection box was used to isolate the sound and Raven automatically generated the following measurements: duration, minimum frequency, maximum frequency, bandwidth and peak frequency. In addition, the start and end frequencies, frequency trend ratio (ratio of the start and end frequencies), frequency range ratio (ratio of the minimum and maximum frequencies), number of inflections and the pulse repetition rate were taken manually (following Dunlop *et al.* [23]). A Classification and Regression Tree (CART) analysis was run to ensure consistent naming of unit types and resulted in a root node error of 93.42% ($n = 1443$) agreement in classification. These measurements, which represent the acoustic features of each unit type, were used to create the weighting system employed in the quantitative analysis (detailed below).

## 2.4. Calculating song and theme similarity using the Levenshtein distance metric

Given the stereotypy in the sequence of units of each phrase type (and therefore theme) and also the sequence of themes that comprised a song type, differences among sequences were striking and could be quantified using common sequence analysis metrics. The Levenshtein Distance (LD) [26] is a robust and powerful edit distance that provides an understanding of song similarity at all levels within the song hierarchy [24,27–28]. The LD calculates the minimum number of alterations required to convert string 'a' into string 'b' [26]. An LD score is calculated by counting the number of insertions, deletions and substitutions and can be normalized to account for the length of the string (the normalized version is referred to as the Levenshtein Distance Similarity Index or LSI). The LSI produces a measure of similarity (between 0 and 1) among multiple sequences of varying lengths and provides an overall understanding of the similarity of all sequences [29]. The LD has been shown to outperform other methods of comparing acoustic sequences by consistently reconstructing the already-known contextual information, for example, clustering humpback whale songs by population [30].

All LSI analyses were run in R [31] using custom-written code (available at https://github.com/ellengarland/leven). Detailed methods for calculating the LD, LSI and applying a weighting system are provided in Garland *et al.* [29]. The LSI was initially run here with all unit sequence data (i.e. the sequence of units making up each phrase, hereafter called a 'phrase string') from every singer and clustered to validate the qualitative assignment of each phrase string to a phrase type (and higher-level theme grouping). Once the theme categories were verified, we conducted two LSI analyses to quantify song similarity at two levels within the song hierarchy. First, the similarity in the sequence of units making up each phrase type (theme) was quantified to allow fine-scale matching among individuals and populations. A representative (median) string was calculated for each individual singer to ensure the typical string of units sang for each phrase type was included. The LSI was run following Garland *et al.* [22] as a weighted analysis where unit substitution costs were calculated based on the acoustic feature similarity between pairs of unit types (quantified as part of CART, described above). Substitution costs were between 0 and 1 depending on the similarity in acoustic feature space between the units, while all other operations (i.e. insertions and deletions) had a cost of 1 (see Garland *et al.* [29] for a detailed account of the method). We used a weighted LSI analysis to allow fine-scale idiosyncrasies to be quantified, such as replacing specific units with a different but similar sound unit within a phrase. These were characteristic to some populations and may be signatures of cultural mechanisms (e.g. convergence on a single song norm, song learning). Second,

the similarity in the sequence of themes making up a song for each individual was quantified to compare broad-scale differences among populations. A representative (median) string of the themes that comprised a song was again calculated per individual singer, and the LSI was run un-weighted.

For both analyses, similarity (LSI) scores were average-linkage (UPGMA) hierarchically clustered and bootstrapped (1000 times) in R using the *hclust*, *pvclust* and *pvrect* packages to ensure the resulting structure was stable and likely to occur [24,29,32]. The approximately unbiased (AU) and bootstrap probability (BP) values, as well as the standard error for each division in the dendrogram were retained. An AU *p*-value of greater than 95% and a BP *p*-value of greater than 70% were considered to be significant, stable and strongly supported by the data [24,32], whereas lower values suggest variability in their division. The clusters deemed stable and supported by the bootstrap analysis (AU *p*-value > 95%) [24] were additionally marked at the highest level with a box using the *pvrect* function [32]. The cophenetic correlation coefficient (CCC) was also calculated as a further, independent test of how well each dendrogram represented the data; scores over 0.8 were considered a good representation of the associations within the data [33].

## 2.5. Alternative test of song similarity and assignment to wintering ground origin

As an alternative and independent test to the LD, we investigated song similarity using the presence of phrase types in each wintering ground compared with each individual singer recorded at the Kermadecs. The presence of all phrase types for each singer recorded at the Kermadecs (KI) and for each wintering ground was noted. The percentage of phrase types from a Kermadec singer that matched the phrase types sung at each wintering ground was calculated by:

$$\% \text{ matched } = \frac{\text{number of shared phrase types with wintering ground}}{\text{Total number of phrase types present in the KI individual's recording}} \times 100.$$

For example, out of the five phrase types sung by Kermadec singer KI01S1, five phrase types matched New Caledonia (therefore 100%), five matched with Tonga (100%), five with Niue (100%) and four with the Cook Islands (80%) (table 1).

The percentage of matched phrase types to each wintering ground was calculated for each Kermadec singer. The wintering ground with the highest percentage was presumed to be the likely wintering ground of origin for the song (table 1). In cases where more than one wintering ground was equally likely (as in the example above), the origin was not specified. The likely wintering ground of origin was also estimated using the LSI at the phrase type/theme level. When the sequence of 'units' that comprised a phrase type from a Kermadec singer was grouped in a stable cluster (AU greater than 95%) with a phrase string from only one wintering ground, that particular wintering ground was suggested as the origin for that song. If a phrase string from a song recorded at the Kermadecs was grouped with multiple wintering grounds or only one phrase type was present, no wintering ground of origin was suggested. This provided a conservative assignment of Kermadec singers to an origin population.

# 3. Results

Three song types were identified in the song recordings from 52 singers collected at 6 wintering grounds across the South Pacific [eastern Australia (*n* = 11), New Caledonia (*n* = 11), Tonga (*n* = 8), Niue (*n* = 7), Cook Islands (*n* = 8) and French Polynesia (*n* = 7)]. Song type 1 was the dominant song in the central Pacific (the Cook Islands and French Polynesia), song type 2 was the most prevalent in the west (New Caledonia, Tonga and Niue) and song type 3 was only recorded in eastern Australia. These songs were compared to 39 'singers' recorded at the Kermadecs (electronic supplementary material, table S1).

## 3.1. Song from the wintering grounds and Kermadecs

The identification of three song types (labelled 1–3) across the South Pacific wintering grounds and the Kermadec migratory stopover in 2015 was supported by both qualitative matching of themes and song types (figure 1) and quantitative LSI classification by hierarchical clustering and bootstrapping of the most representative song (median string) from each individual singer (figure 2). There were two distinct versions of song type 1, '1a' and '1b', based on phrase type (theme) presence (1a themes: 7,8,9,10,11 and 1b themes: 7,9,11,12,13,14). Hierarchical clustering and bootstrapping supported this

**Table 1.** Likely wintering ground origin for each singer recorded at the Kermadec Islands. The likely origin was determined using a combination of the proportion of phrases present which matched each wintering location (NC, New Caledonia; TO, Tonga; NI, Niue; CI, Cook Islands, FP, French Polynesia; EA, east Australia) and the LSI similarity analyses. The likely origin for the song was determined when one wintering ground had a higher percentage of phrases matched than the other wintering grounds. The LSI similarity origin was determined using both the fine-scale phrase level and broad-scale song-level cluster analyses: when the median phrase string from a Kermadec Islands singer clustered (AU p-value >95%) with a string from a single wintering ground (no multi-origin clusters were included in the table). The likely origin was determined when a singer from the Kermadec Islands was linked to one wintering ground on fine or broad-scale analysis. Kermadec singers with a single phrase type (n = 8) were excluded from the analysis and were not assigned to a breeding ground.

| Kermadec singer # | total # phrases | % matched phrases | | | | | | | | | LSI similarity assignment | | | | | | likely origin | | |
|---|---|---|---|---|---|---|---|---|---|---|---|---|---|---|---|---|---|---|---|
| | | song 1 | | | | song 2 | | | | song 3 | theme | | | | | | | LSI | |
| | | NC | TO | CI | FP | NC | TO | NI | CI | EA | 1 | 2 | 3 | 5 | 6 | 14 | % | phrase | song |
| KI01S1 | 5 | 0 | 0 | 0 | 0 | 100 | 100 | 100 | 80 | 0 | TO | | | | | | | TO | TO |
| KI01S1a | 8 | 0 | 0 | 0 | 0 | 100 | 87.5 | 87.5 | 62.5 | 0 | | TO | | | | | NC | TO | TO |
| KI01S1b | 5 | 0 | 0 | 0 | 0 | 100 | 100 | 100 | 80 | 0 | TO | | TO | | | | | TO | |
| KI02S1a | 5 | 0 | 0 | 0 | 0 | 100 | 80 | 80 | 60 | 0 | | | | | | | NC | | |
| KI02S2 | 2 | 0 | 0 | 0 | 0 | 100 | 100 | 50 | 50 | 0 | | | TO | | | | | TO | |
| KI02S3 | 5 | 0 | 0 | 0 | 0 | 100 | 100 | 100 | 60 | 0 | | NC | | | NC | | | NC | |
| KI03S3 | 5 | 0 | 0 | 0 | 0 | 100 | 100 | 100 | 60 | 0 | | TO | TO | | TO | | | TO | |
| KI03S4[a] | 7 | 0 | 0 | 0 | 0 | 85.7 | 85.7 | 85.7 | 42.9 | 0 | | | | | | | | | NI |
| KI03S5 | 2 | 0 | 0 | 0 | 0 | 100 | 100 | 50 | 50 | 0 | | | | | | | | | |
| KI04S1[a] | 3 | 0 | 0 | 0 | 0 | 66.7 | 66.7 | 66.7 | 66.7 | 0 | | | | | | | | | |
| KI04S1b | 4 | 0 | 0 | 0 | 0 | 75 | 100 | 100 | 75 | 0 | | | | | | | | | |
| KI04S2 | 4 | 0 | 0 | 0 | 0 | 100 | 100 | 100 | 100 | 0 | TO | | | | | | | TO | |
| KI04S3 | 5 | 0 | 0 | 0 | 0 | 80 | 80 | 100 | 40 | 0 | | | | | | | NI | | |
| KI05S1 | 6 | 0 | 0 | 0 | 0 | 83.3 | 83.3 | 100 | 50 | 0 | | | | | TO | | NI | TO | NC |
| KI05S2[a] | 11 | 0 | 0 | 0 | 0 | 90.9 | 81.8 | 90.9 | 54.6 | 0 | | NC | TO | | | | | TO | |
| KI05S3 | 5 | 0 | 0 | 0 | 0 | 100 | 100 | 100 | 80 | 0 | | | | | TO | | | TO | |

**Table 1.** (*Continued.*)

| Kermadec singer # | total # phrases | % matched phrases | | | | | | | | song 3 | LSI similarity assignment | | | | | | likely origin | | |
|---|---|---|---|---|---|---|---|---|---|---|---|---|---|---|---|---|---|---|---|
| | | song 1 | | | | song 2 | | | | | theme | | | | | | LSI | | |
| | | NC | TO | CI | FP | NC | TO | NI | CI | EA | 1 | 2 | 3 | 5 | 6 | 14 | % | phrase | song |
| KI06S1 | 10 | 0 | 0 | 0 | 0 | 100 | 80 | 90 | 50 | 0 | TO | NC | | | NC | NC | NC | | |
| KI10S1 | 5 | 0 | 0 | 0 | 0 | 100 | 80 | 100 | 40 | 0 | | | | | | | | | |
| KI10S2b | 3 | 0 | 0 | 0 | 0 | 100 | 100 | 100 | 100 | 0 | | | | | | | | | |
| KI11S1 | 2 | 0 | 0 | 0 | 0 | 100 | 100 | 100 | 50 | 0 | | | | TO | | | | TO | |
| KI12S1 | 3 | 0 | 0 | 0 | 0 | 100 | 100 | 100 | 66.7 | 0 | | TO | TO | | | | | TO | |
| KI12S2 | 5 | 0 | 0 | 0 | 0 | 100 | 80 | 80 | 80 | 0 | | | | | NI | | NC | TO | |
| KI12S3 | 7 | 0 | 0 | 0 | 0 | 100 | 85.7 | 85.7 | 71.4 | 0 | TO | | | | NC | | NC | | |
| KI12S4 | 10 | 0 | 0 | 0 | 0 | 100 | 90 | 90 | 60 | 0 | | | NC | NI | CI | | NC | | |
| KI12S5 | 10 | 0 | 0 | 0 | 0 | 100 | 80 | 90 | 60 | 0 | TO | | | | | | NC | TO | |
| KI12S6 | 6 | 0 | 0 | 0 | 0 | 100 | 66.7 | 66.7 | 50 | 0 | | | NC | | NC | | NC | NC | |
| KI12S7 | 5 | 0 | 0 | 0 | 0 | 100 | 80 | 100 | 40 | 0 | | | | | | | | | |
| KI13S1 | 11 | 0 | 0 | 0 | 0 | 100 | 81.8 | 90.9 | 63.6 | 0 | | | | | CI | | NC | CI | |
| KI14S1 | 10 | 0 | 0 | 0 | 0 | 100 | 80 | 90 | 60 | 0 | NI | | | | NC | | NC | | NC |
| KI14S1a | 7 | 0 | 0 | 0 | 0 | 83.3 | 100 | 100 | 83.3 | 0 | TO | NI | | | | | CI | CI | |
| KI14S2 | 3 | 66.7 | 0 | 100 | 33.3 | 0 | 0 | 0 | 0 | 0 | | | | | | CI | CI | CI | |

aIndicates a singer found to be performing a phrase exclusively found in New Caledonia *and* phrases which were not present in the New Caledonian recordings. See §3.2 Song transcription, for designation of 'individual' Kermadec singers and electronic supplementary material, table S2 for further information on singers.

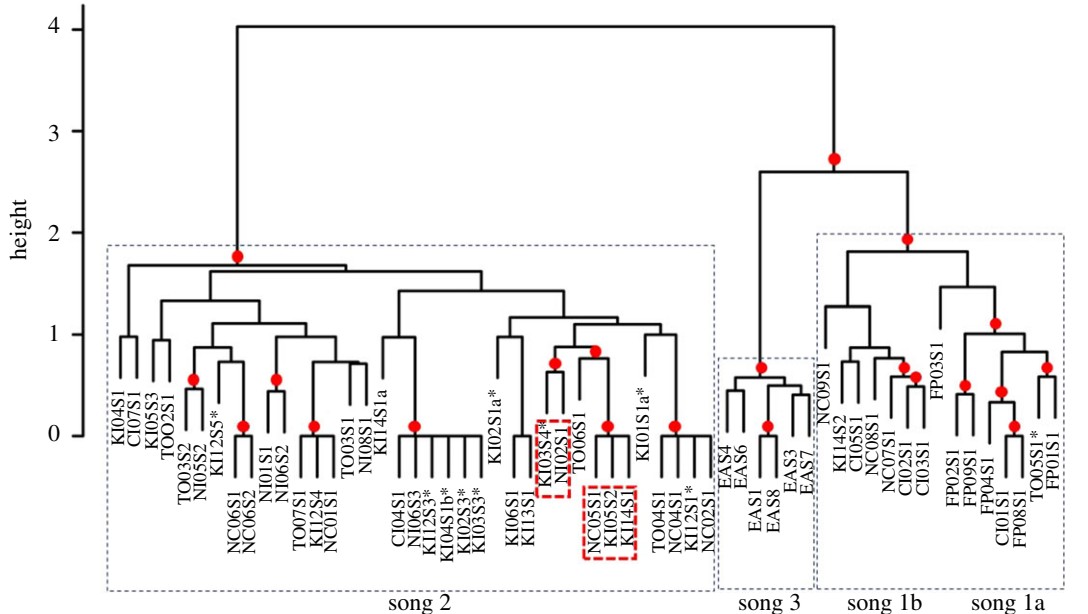

**Figure 2.** Dendrogram of the similarity between the representative song (theme sequence) for each individual singer recorded at the Kermadec Islands (KI) and at each of the six western and central South Pacific wintering grounds (EA, eastern Australia; NC, New Caledonia; TO, Tonga; NI, Niue; CI, Cook Islands; FP, French Polynesia). The median string LSI scores were hierarchically clustered using average-linkage clustering and bootstrapped ($n = 1000$). The AU values (significant $p$-values >95%, red dot [24,32]) indicated the structure and divisions in the tree were stable and likely to occur. This was additionally confirmed using the Cophenetic Correlation Coefficient, which indicated that the structure of the tree was a very good representation of the associations present within the data (CCC = 0.97). Black dashed boxes delineate each song type. Red dashed boxes highlight where a singer from the Kermadecs has been linked to a wintering ground within a stable cluster. Singer name is created from the wintering ground, recording number and singer number. For example, singer code NC05S1 is New Caledonian recording number five, singer number one. * indicates an incomplete song sequence from the Kermadecs, although only whales recorded singing more than one theme were included.

division of song type 1 into *a* and *b* versions (figure 2; AU $p$-value = 100%, BP $p$-value = 98%, SE < 0.01). One (out of 8) singer from Tonga, one (out of 8) singer from the Cook Islands and all singers recorded in French Polynesia ($n = 7$) sang song type 1a. Song type 1a was not present in recordings from the other three wintering grounds or at the Kermadecs. Song type 1b was recorded only at the start of the season in New Caledonia (3 out of 11 singers; electronic supplementary material, figure S2), while it was present in later months in the Cook Islands (3 out of 8 singers). Two (out of 39) singers in the Kermadecs were recorded singing song type 1b; however, one singer was only recorded singing theme 14 and therefore was not included in song-level analyses. All singers from Niue ($n = 7$) sang song type 2, while 8 (out of 11) singers from New Caledonia, 7 (out of 8) singers from Tonga and 4 (out of 8) singers from the Cook Islands were also recorded singing song type 2. Qualitative and quantitative analysis showed each population sang a particular, population-specific fine-scale combination of the phrase types, allowing each to be acoustically identified based on this combination. Song type 2 was also the most commonly recorded song at the Kermadecs (36 out of 39 singers). One singer from the Kermadecs sang themes from song type 2 along with a phrase variant of one theme from song type 1a (i.e. hybrid singer, defined as a singer who sings a song containing themes from two song types [6,11]). Finally, song type 3 was only recorded in eastern Australia (figure 2; electronic supplementary material, table S1).

## 3.2. Linking Kermadec singers to a possible origin

Certain phrase types were present at one wintering ground (population) and not recorded at others (electronic supplementary material, table S1 and results). The Kermadec recordings contained 22 (of 38) phrase types; these included all of the phrase types present in song type 2 and three from song type 1b, while a single phrase variant from song type 1a (8C, used only by the hybrid singer) was only recorded in the Kermadecs. The presence/absence of phrase types in each wintering ground as a

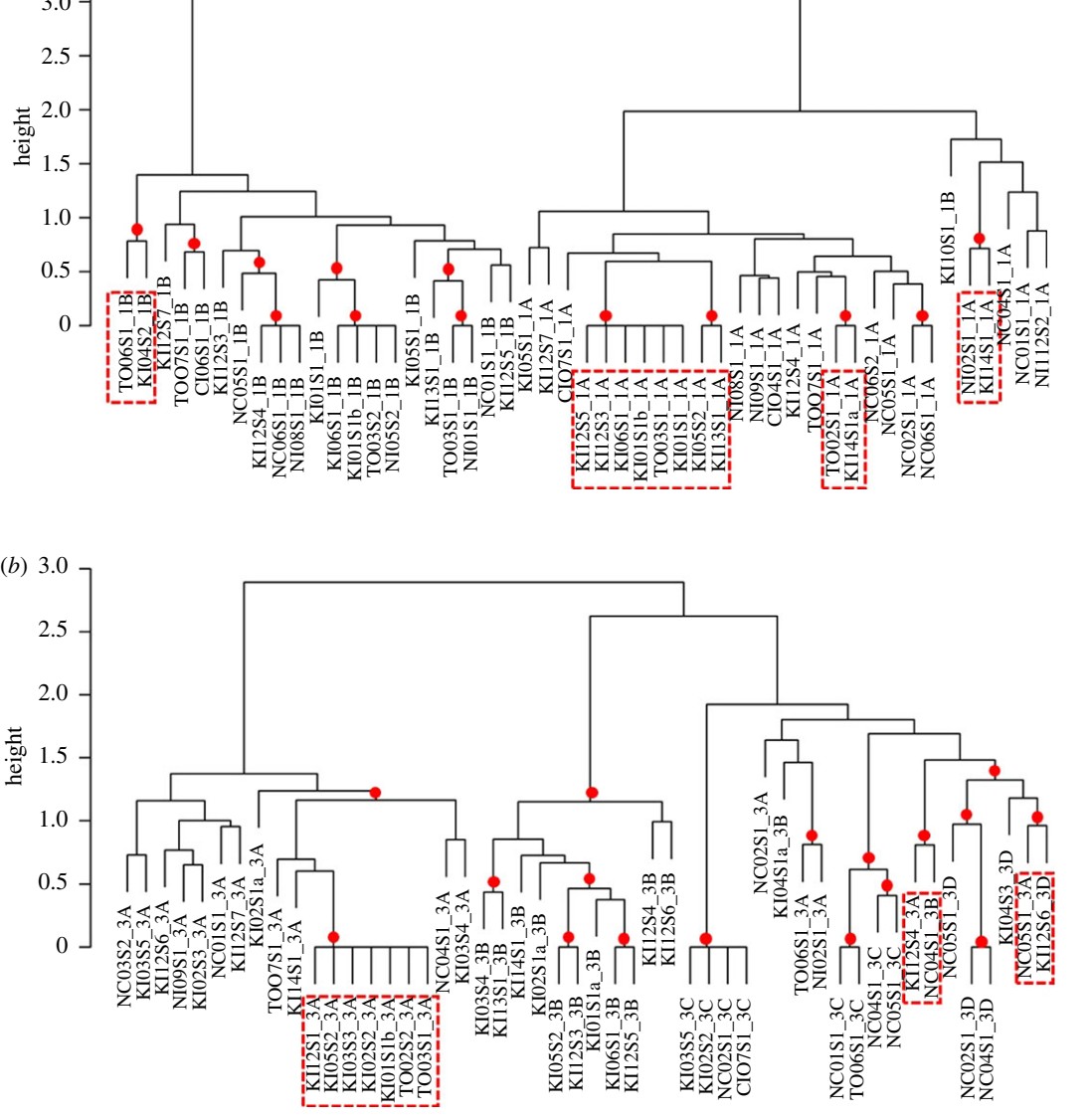

**Figure 3.** Dendrograms representing the similarity of the median sequence of units in (*a*) theme 1 and (*b*) theme 3 from song type 2 for each individual singer recorded at the Kermadec Islands (KI) and the four wintering grounds in which these themes were present (NC, New Caledonia; TO, Tonga; NI, Niue; CI, Cook Islands). Each theme was split into stereotyped phrase types (theme 1: 1A and 1B; theme 3: 3A, 3B, 3C and 3D). The median string LSI scores were hierarchically clustered using average-linkage clustering and bootstrapped ($n = 1000$). The AU values (significant *p*-values >95%, red dot [24,32]) indicated the stability of each split in the tree. This was additionally confirmed using the Cophenetic Correlation Coefficient, which indicated that the structure of both trees was a very good representation of the associations present within the data [(*a*) theme 1 CCC = 0.96, (*b*) theme 3 CCC = 0.94]. Red dashed boxes highlight where a singer from the Kermadecs has been linked to a wintering ground within a stable cluster. Singer name is created from the wintering ground, recording number, singer number and theme name. For example, singer code KI12S2_1A is Kermadecs recording number 12, singer number two, theme 1A.

whole was compared with each individual singer recorded at the Kermadecs to calculate the percentage of matched phrase types. Three singers at the Kermadecs sang a phrase type recorded only in New Caledonia (phrase 3B) *and* one or more phrase types that had not been recorded in New Caledonia. For consistency, these three singers and any other singers from the Kermadecs who were only recorded singing one theme type were not included in this part of the analysis. The percentage of matched phrases suggested a wintering ground of origin for 13 out of 26 singers at the Kermadecs: 10 from New Caledonia, 2 from Niue and 1 from the Cook Islands (table 1). The remaining 13 singers matched multiple wintering grounds and were thus left as unassigned.

Using the most representative song (i.e. composed of a sequence of themes) from each individual singer, the LSI song scores were clustered. If a stable cluster (AU $p$-value > 95%; represented by a red dot, figure 2) contained a Kermadecs singer with just one wintering ground, a likely origin population was determined. Within song type 2, stable clusters linked two Kermadec singers with New Caledonia and one Kermadec singer with Niue (red boxes, figure 2). The remaining Kermadec singers within the song type 2 cluster were part of multi-origin clusters, although an east–west divide was evident (New Caledonia and Tonga versus Niue and Cook Islands) in all remaining stable Kermadec and wintering ground groups (figure 2). Of the two Kermadec song type 1b singers, one singer was consistently grouped with song sequences from the Cook Islands, although this was not quite a stable cluster (AU = 89, figure 2). The second singer was represented by a single theme [14] and was therefore excluded from figure 2; however, fine-scale analysis linked this Kermadec recording with the Cook Islands (see below; electronic supplementary material, figure S1).

To assess the fine-scale (phrase type) similarity between songs recorded at the Kermadecs and each wintering ground, a representative (median) unit sequence for each phrase type per singer was compared for each theme. Hierarchical clustering and bootstrapping of the similarity scores for all individuals recorded singing themes 1, 2, 3, 5, 6 and 14 linked singers from the Kermadecs to New Caledonia, Tonga, Niue and the Cook Islands (table 1, figure 3a,b; electronic supplementary material, figure S1). Themes 4, 7, 8 and 12 produced a varied picture of phrase type origin as Kermadec singers were placed with singers from multiple wintering grounds. The remaining themes present at the wintering grounds (9, 10, 11, 13, 15, 16, 17) were not recorded in any of the Kermadec songs.

In summary, the percentage matched phrase analyses suggested that 10 Kermadec singers were likely to have originated from New Caledonia, 2 from Niue and 1 from the Cook Islands, whereas the fine-scale (phrase) LSI analyses suggested that 11 of the Kermadec singers were from Tonga, 2 from New Caledonia and 2 from the Cook Islands (table 1). The broad-scale (song) LSI analysis suggested two Kermadec singers originated from New Caledonia and one from Niue (table 1). Combining all lines of evidence, there were three occasions where all analyses agreed on the likely wintering ground of origin for a Kermadec singer: KI12S6 and KI14S1 were assigned to New Caledonia, and KI14S2 was assigned to the Cook Islands (table 1). All other singers' origins were left unassigned due to a lack of agreement among analyses.

# 4. Discussion

Whales from many different populations were passing the Kermadecs at the same time (table 1), providing the opportunity for song learning and easterly transmission of song across the South Pacific. This is clearly one of potentially multiple important locations for the cultural transmission of humpback song. Owing to fine-scale differences identified in the song recordings from the wintering grounds analysed here, it was possible to match the songs recorded at the Kermadecs to New Caledonia, Tonga, Niue and the Cook Islands, a pattern mirrored by data on genetically and photographically identified individual whales in the same area [21]. These song analyses also suggested that singers from both French Polynesia and eastern Australia were unlikely to visit this migratory location. In general, little is known about the migratory routes of South Pacific humpback whales. The temporary aggregation of whales at a migratory stopover may be related to humpback whales' strong urge to socially aggregate [34] and/or due to following oceanographic landmarks on migration [35]. Regardless of the underlying driver, the stopover provides an opportunity for acoustic connectivity among multiple migratory streams. The hybrid song recorded at the Kermadecs is consistent with the hypothesis of song learning on migration [6,11], and although such an aggregation of whales at a migratory stopover is temporary, it may be a major driver facilitating the easterly transmission of song across the South Pacific. Furthermore, as humpback whales sing less frequently on the feeding grounds [19], and as far as we know, do not aggregate at the Kermadecs during the northward migration, the song learned during the southward migration needs to be remembered until the next breeding season (akin to song memory in oscine songbirds [36]).

## 4.1. Possible origins of Kermadec singers

Analyses suggested the songs recorded at the Kermadecs matched all sampled South Pacific wintering grounds except eastern Australia and French Polynesia, despite photo-identification and genetic studies demonstrating the occasional between-season movement of individuals and thus connectivity

across the wider South Pacific region [37–39]. A concurrent genetic and photo-identification study from the Kermadecs [21] strongly reinforced our findings: individuals from multiple populations (excluding eastern Australia and French Polynesia) were passing the Kermadecs on their southward migration. Thus far, genetic and satellite tagging studies have not found a migratory link or route between eastern Australia and the Kermadecs [40–42], and although the exact location of the summer feeding ground(s) or the migratory routes for the French Polynesian population remains elusive, photo-identification and genetic matches have been found with the Antarctic Peninsula (South America) to the east [43]. Migrating via the Kermadecs would represent a large and potentially unnecessary deviation for both eastern Australian and French Polynesian whales.

## 4.2. Acoustic contact allows easterly transmission

Owing to the consistent easterly transmission of song types across the South Pacific [9,18–19], multiple song types were identified. The low frequency of song type 1 (the 'oldest' song type) in the recordings from the western South Pacific region follows previous patterns [9], as the 'newer' song type 2 from the west was rapidly replacing it. Song type 1b was only recorded early in the season in New Caledonia prior to singers switching to the 'new' song type 2 (electronic supplementary material, figure S2), while it was recorded throughout the season at the Cook Islands. As expected, there was also a low frequency of song type 1 at the Kermadecs. The strong pairing of themes (figure 3) from the two whales singing song type 1b, indicated that both these Kermadec songs (and thus singers) likely originated from the Cook Islands. This complements evidence from photo-identification and genetic matches [21], that some whales from the Cook Islands, a central South Pacific 'population', pass the Kermadecs on their southwardly migration. It is still unclear, however, whether the Cook Islands represents a migratory corridor or a distinct winter breeding ground [39,44]. Regardless, whales sampled in the Cook Islands are genetically most similar to those wintering in Tonga, but in previous studies, acoustically more similar to those from French Polynesia [9,18,20,39,44]. Central South Pacific populations (the Cook Islands and French Polynesia) need to come into contact at some point in the year with whales from the western group (e.g. New Caledonia or Tonga) to allow cultural transmission of the song. The Kermadecs provide a location where eastern whales may be exposed to new song types from western populations, emphasizing the importance of shared migratory routes or stopovers for the cultural transmission of song.

## 4.3. Song learning and potential for convergence

One song recorded at the Kermadecs mostly contained song type 2 themes but also one theme from song type 1 (i.e. a hybrid song). This follows the pattern observed in the wintering grounds, where whales in the western South Pacific region have switched, or are in the process of switching, to song type 2. Hybrid songs are rare and likely short lived [6,11], so this hybrid song, with which we have likely captured some part of the process by which singers change their song display from an older to a new song version, suggests that the Kermadecs are a location where song learning occurs. Whales stay on average 4.6 days at Raoul Island (R.C. unpublished data, 2015), which potentially allows for exposure to song for several days. Such song exposure may allow both the learning and transmission of population-specific phrase types among populations and acquisition of new songs facilitating the easterly transmission of these songs across the South Pacific. While this addresses how songs may spread east, the question of why songs only spread in an easterly direction remains unresolved. One possible explanation is the difference in population sizes; the large eastern Australian population may influence the smaller South Pacific populations more then they influence it [9]. Future research examining population size, density and transmission, potentially using agent-based models [45], may help to answer this question.

On the wintering grounds, South Pacific humpback whales are acoustically isolated (song only travels effectively for tens of kilometres at most [46]) from other wintering grounds due to their natal site fidelity. Previous South Pacific song studies have noted population-level differences in phrase type presence [20]. Song from three Kermadec singers contained a phrase type which was sampled only from New Caledonia *and* one or more phrases which were not identified in New Caledonian recordings during the preceding season. During temporary acoustic isolation at the separate wintering grounds, progressive song evolution creates fine-scale population differences [16,20]. The fine-scale LSI analysis is consistent with the hypothesis that the reestablishment of acoustic contact at the Kermadecs may have initiated the merging of these population-level differences in song (type 2) to a single norm, resulting in a degree of cultural convergence.

Conforming to a cultural norm is important in many taxa [47]. In humans, conformity stems from a motivation to copy others to fit into a social norm [48]; children will abandon behavioural preferences in favour of imitating peers [49]. Cultural boundaries, often created by language differences, give humans a means of establishing who to socialize and cooperate with [50]. Vocal conformity and dialect boundaries are both prominent features of humpback whale song, and these features are also coupled with a constant evolution of the trait. How cultural evolution and sexual selection each contribute to this cultural phenomenon remains an open question.

# 5. Conclusion

Here, our results are consistent with the conclusion that song learning, transmission and potentially convergence occurs at the Kermadecs, where migratory routes from multiple populations overlap. This contact is likely to be a driver of horizontal transmission of song across the South Pacific. While convergence and transmission have been shown within a population during migration and on their wintering grounds [9,11], song exchange and convergence on a shared migratory route, and the location of such an event, remained elusive. Song themes from multiple wintering grounds matched songs recorded at the Kermadecs, including a hybrid of two songs, suggesting that multiple humpback whale populations from across the South Pacific are travelling past these islands and song learning may be occurring. Although it is not possible to discount other potential mechanisms of song transmission [16], our results are consistent with the hypothesis of song learning on a shared migratory route, a mechanism that could drive the eastern transmission of song across the South Pacific [9].

Ethics. The University of St. Andrews School of Biology Ethics Committee approved this study.

Data accessibility. The datasets supporting this article (comprising raw song transcripts and unit measurements) have been uploaded as part of the electronic supplementary material.

Authors' contributions. C.O. transcribed acoustic samples, conducted data analysis and drafted the manuscript (with supervision from E.C.G. and L.R.); E.C.G. conceived and designed the study, contributed acoustic data, supervised data analysis, assisted with interpretation of results and critically revised the manuscript; L.R. assisted with interpretation of results and critically revised the manuscript; R.C. conceived and designed the study, contributed acoustic data and revised the manuscript; M.J.N. conceived and designed the study, contributed acoustic data and revised the manuscript; J.A., C.G., D.D., O.A., N.H. and M.M.P. contributed acoustic data and revised the manuscript. All authors contributed to the editing of the manuscript and were responsible for the approval of the final manuscript.

Competing interests. We declare we have no competing interests.

Funding. C.O. was partially supported by the Sidney Perry Foundation, and the NERC Sea Mammal Research Unit made a contribution towards the write up of this study. E.C.G. was supported by a Royal Society Newton International Fellowship and a Royal Society University Research Fellowship. L.R. was supported by the MASTS pooling initiative (The Marine Alliance for Science and Technology for Scotland) and their support is gratefully acknowledged. MASTS is funded by the Scottish Funding Council (grant reference HR09011) and contributing institutions. J.A. was supported by an Australian Government Research Training Program Scholarship, Australian American Association University of Queensland Fellowship and the Sea World Research and Rescue Foundation Inc. Research in the Kermadec Islands was primarily supported by the New Zealand Ministry for Primary Industries BRAG Fund, a University of Auckland FRDF Grant, the Australian Antarctic Division, and the Pew Charitable Trusts. Data collection in eastern Australia was funded by the E&P Sound and Marine Life Joint Industry Program (JIP) and the US Bureau of Ocean Energy Management as part of the BRAHSS project. M.M.P. was partially supported by the National Oceanic Society and Dolphin & Whale Watching Expeditions.

Acknowledgements. We thank Blue Planet Marine, Conservation International, the crew of the *RV Braveheart*, the research team—Rémi Dodémont, Becky Lindsay, James Tremlett, the Raoulies and our colleagues at the South Pacific Whale Research Consortium, for supporting research in the Kermadec Islands. Many thanks to Ngāti Kuri and Te Aupōuri for allowing us to work with their taonga. Research was conducted under a Department of Conservation Permit #44388-MAR to R.C. The BRAHSS project thanks Rebecca Dunlop, Doug Cato and the numerous staff and volunteers who have assisted with acoustic data collection. Thanks to Patrice Plichon and the Province Sud, as well as Véronique Perard, Rémi Dodémont and Solene Derville from Operation Cétacés for their help with data collection in New Caledonia. The Niue Whale Research Project thanks our project partner Fiafia Rex of Oma Tafuà for collecting song data and without whom this work could not happen in Niue. Cook Islands research would like to thank Joan Hauser Daeschler, Rose Cottage Outreach, Alyssa Stoller and Team. D.D. would like to thank Blue Water Explorer Ltd. Fieldwork in French Polynesia was conducted under a permit issued to M.M.P. from French Polynesia's Ministry of the Environment. The authors thank Patrick Miller and two anonymous reviewers for comments that improved previous versions of this manuscript.

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
