## [Reviewer comments · Royal Society Open Science]

Review History

RSOS-190337.R0 (Original submission)

Review form: Reviewer 1 (Christopher W. Clark)

Is the manuscript scientifically sound in its present form?

Yes

Are the interpretations and conclusions justified by the results?

No

Is the language acceptable?

Yes

Do you have any ethical concerns with this paper?

No

Have you any concerns about statistical analyses in this paper?

No

Recommendation?

Accept with minor revision (please list in comments)

Comments to the Author(s)

Reviewer Comments: Owen-etal_Humps_RSOS-2019

Title might be refined as it seems a bit inverted. Doesn't transmission of song occur before "cultural convergence"? How do you know it's simultaneous given the very limited period of time during which acoustic samples were collected and the sparse sampling locations over a vast ocean area? Title simplified might better read: "Migratory convergence facilitates transmission and cultural convergence of humpback whale song"?

Summary seems quite selective in terms of cultural transmission: vertebrates, humans, cetaceans. I suggest a very careful review of the wording for detailed accuracy and revise as needed.

Examples:

Line 7: Given that "birds to humans" implies only vertebrate species are considered to engage in cultural transmission behaviors, I suggest inserting "vertebrate" before "taxa".

Line 9: Consider changing "sing" to "perform" so as to read " perform complex, culturally transmitted song displays that can change in ... " The way this is presently stated, it reads as if all populations change songs in both ways, when this has not yet been demonstrated in all populations.

Line 12: Areas of migratory contact remain unknown ... "across vast oceanic regions". This is an example of unnecessary hyperbole. "these areas of contact remain unknown" - Isn't it more the case that the specific areas of acoustic contact remain uncertain? Look at tagging results that show general, broad migratory "avenues" in other "vast" ocean regions - e.g. North Pacific and North Atlantic.

Line 13: "Here, we linked song recordings from..." You actually did not "link" song recordings (Recording is a mechanism, and what does "link" mean?). Isn't it more accurate to say that you identified songs recorded from the Kermadec Island area during spring migration as having similarities to songs recorded from a handful of South Pacific island areas during the winter? You then interpreted such similarities as evidence of song transmission between populations.

Lines 15-17: Sentence would benefit from a rewrite; in its present form it's a bit convoluted and unclear. Delete "in recordings" as this is assumed. Songs don't demonstrate a mechanism, though they might imply one or are a manifestation of one. Songs are a behavior that you recorded and analyzed for similarities, and from those similarities you derived some conclusions: similarities in song themes from the Kermadec Island area and South Pacific island areas indicate/suggest/are consistent with the conclusion that

Introduction:

Various, minor grammatical edits needed; for example, commas between multiple adjectives.

Lines 43-44: Whale populations don't "pass songs" onto another populations. Isn't it more appropriate to say that individuals in one population hear songs from another population and adopt song components into their repertoire.

Lines 54-55: Based on the literature and my own experiences, I would agree that songs are most frequently produced and recorded on winter breeding grounds, but less frequently while whales are migrating.

Line 59: “pinpointing” is jargon. Why not simply use “identifying”?

Lines 70-72: I suggest merging the first and second sentences so as to read something along the lines: “Here we hypothesised that if individual males migrated into the Kermadecs from multiple wintering sites during their southward migration (September and October), we should see some evidence of the cultural processes, song transmission or convergence.”

Section 3.1, Song Recordings (Lines 85-103): Section needs some further info on recording conditions (e.g. depth of hydrophone, depth of water, sound velocity profile) to help reader better understand the range of song detection as a function of frequency and quality of recordings (e.g. low-frequency stripping). Also, last sentence seems to contradict previous sentences - that there was not a common sampling rate. For spectrogram display please state time, frequency and grey-scale level resolutions. Somewhere the text would benefit from discuss and deal with the issue of range of detection for singers.

Text says that Kermadecs’ acoustic data consisted of 12 recording sessions throughout 13 days, but Table S1 lists 7 days, 2-8 October 2015. Please clarify. The total amount of acoustic recordings for all sites was only 19 hours and 9.5 minutes yielding song material from an estimated 91 singer. That’s an average of only 12.6 mins of song per singer, which means your data does NOT incorporate song variability. You must underscore this very low sampling of song variance in your Abstract, Summary, Discussion and Conclusion.

Section 3.2 – 3.5: Song transcription (Lines 106-119): Song transcription process includes a certain level of subjectivity, which needs to be clearly stated. Kermadec Island recordings yielded 39 “individual singers” at unit-level and 17 at theme level from 12 recordings.

What’s a “phrase string” – same as a song?

The supplemental acoustic song samples are only about a minute long, so one cannot conclude very much about auditory song similarity and variability from such short samples. Table 1 needs to list the duration of the singing for each “singer #”. How does one know whether the same singer contributed more than once to the data listed in Table 1?

What’s a “hybrid singer”? = song composed from phrase types recorded from singers recorded at multiple wintering sites?

How do authors rectify-integrate delay in a whale’s acquisition of new song material with his actual singing performance that includes that new material? Is it assumed that individuals assimilate novel songs into singing behavior immediately?

This paper is based on a very small snippet, an auditory wink-blink into this phenomenon of song sharing, acquisition, and “acoustic polymorphism.” The authors must make this critical fact (of a very limited data set) absolutely clear from the beginning of the manuscript, especially given that they write as if they have sampled a “vast” expanse of the South Pacific, when in fact they have not. Sampling was sparse over time and space and thus potentially subject to strong levels of aliasing depending on one’s level of analysis and interpretation.

Discussion

Line 268: Why is it appropriate to declare Kermadecs a significant location? Isn't this the first time that this type of song comparison has been done in this region? Why wouldn't this phenomenon be occurring all over the South Pacific, or is it assumed to occur only around islands? Aren't you recording around islands because it's easier and logistically feasible? The paper needs to adequately place these intriguing, initial results within the broader context of the South Pacific region and not write as if these are the final chapter, when in fact this is really the preface.

Line 270: The authors often take liberty to make these types of sweeping statements about the vast distances over which samples were taken so as to make it seem as if they sampled throughout that expanse, when in fact they did not! 5-6 sample sites across a 6000 km expanse of ocean is, by definition spatially aliased. So, please pull back a bit on the over exuberance of these preliminary findings, and change your focal depth such that you interpret these insights as very early discoveries into a vastly complex phenomenon. Otherwise, what words will you have left to use when the even greater discoveries emerge as you all brush off the next layer in this geographical, acoustic expedition.

Line 281 – Yes, finally, this is an important point, but you never develop it! Why not make the obvious parallel link to oscine songbirds that will sing novel song material many months after exposure to that new material? Acoustic memory and song acquisition is not new in the universe of animal song.

Section 5.1, Lines 283-293: It is not surprising that not all wintering songs were detected in such a small sample over such a short period of time! The reliance to Riekkola et al. 2018 suffers the same problem since samples in that paper were only acquired during 12 days in 2015.

This entire puzzle is intriguing but suffers from spatial and temporal aliasing (Yes, I'm being redundant!) that needs to be explicitly recognized. This would not necessarily detract from the import of this study, while also appropriately placing its results in a much more honest and appropriate scientific context.

Section 5.3, Lines 314-338: "captured a whale in the process of learning a new song display" = really worded like this? How do you know that this whale was in the process of learning a new song when you have only a crumb of recording for this animal? You, don't, so you cannot make such a definitive statement. The self-recognition refs (6 and 11) do not really support this statement. Again, claiming that these few observations from this location are evidence that the Kermadecs are a significant location is premature, and such self-centered awards are an unnecessary distraction for a reader. Let the reader decide if these are "significant" – whatever that means in the broader context of this paper. Instead – why does the song move east? Magnetism? Geothermal? Oceanographics?

It would be extremely valuable if the authors would come up with a better way of illustrating the Spatio-temporal dynamics of this phenomenon. The Figure 1 illustration is helpful, but static. None of the illustrations are really novel, and thus they are not an effective means by which to allow your audience to visualize, and thus readily comprehend the dynamics of the cultural song-learning, song-sharing phenomenon through time and space.

Lines 327-328: The LSI analysis did not "indicate" this. It "suggested" something that is consistent with a hypothesis at the authors might want to interpret as an indication, but using "indicate" is not supported by the very scant data or their analysis. This text is another example where the authors go beyond the boundaries and overinterpret their results. Unacceptable!

Lines 335-338: "Given that humpback [6],... culture" This text is really too much! There is no

direct evidence, just pure speculation, that humpbacks have any of the standard mechanisms for human speech or oscine songbird song acquisition. Implying humpback song research as a way of exploring human language is preposterous as well as amazingly naïve; at best it implies that the authors are unfamiliar with those deep bodies of scientific discovery. This text is totally a distraction, unnecessary and a diversion from the results of this paper, and the authors have done nothing to support such a claim except reference a previous interesting paper that superficially speculates.

Note, line 332, “boundaries” is plural, so “gives” needs to be “give.”

Lines 342-350: The first sentence is another misstatement and overstatement. You do not know anything about what any of the singers you recorded at the Kermadecs were singing before or after your brief song samples, or what songs they were exposed to. You only know what they sang when you recorded them. Therefore, you do not have enough data, no matter how complex the analysis, to state that you have demonstrated that convergence and/or song learning occurred at the Kermadecs. Rather, you have shown that your results are consistent with a conclusion/hypothesis that these occurred. Stating your results correctly does not diminish their importance, and you should be satisfied with these preliminary insights. Overstating your results is neither wise or acceptable.

What’s evidence have you provided that indicates “song evolution”? How can you claim this from such a limited sample. How do you know that songs “originated” on a wintering ground from these data? This all sounds very circular! All you know is that you recorded songs that were similar to those recorded someplace else. Since the whales are able to acquire songs, you don’t know that songs recorded somewhere else originated there. If animals are exposed to a song type in area A, but wait to sing that song many months later in area B; what’s the chicken and what’s the egg.

Line 347: I argue that “many” should be “multiple”. Since you do not have an adequate time series of samples from the individual singer at the same location throughout the song learning period, you cannot state that song learning is occurring at the Kermadecs site; you can only state that singers were recorded singing songs that shared characteristics of songs recorded at other sites. You can state that these results are consistent with the hypothesis that

This last comment can be applied throughout this paper, which is full of over-interpretations from the very limited data set. It’s an important observation, but cannot claim all these claims!!

Review form: Reviewer 2

Is the manuscript scientifically sound in its present form?

Yes

Are the interpretations and conclusions justified by the results?

Yes

Is the language acceptable?

Yes

Do you have any ethical concerns with this paper?

No

Have you any concerns about statistical analyses in this paper?

No

Recommendation?

Accept as is

Comments to the Author(s)

I think this is good paper which continues on from previous findings of Garland et al. The methodology described in the paper to measure cultural transmission of Humpback Song seems to be a robust, and it would be good if it was adopted by other researchers covering other regions. The only issue I had was really minor to do with the color selection in Figure 1. The blue/greens used for song types 1a, 1b and 2 are very close and it is hard to discriminate them in the ellipses surrounding the islands. I would also be interested in how song complexity plays a role as well as to why cultural transmission only seems to travel from West to East is it just to do with relative stock size or is there something else occurring?

Decision letter (RSOS-190337.R0)

17-Jul-2019

Dear Dr Garland

On behalf of the Editors, I am pleased to inform you that your Manuscript RSOS-190337 entitled "Migratory convergence allows simultaneous cultural convergence and transmission of whale song" has been accepted for publication in Royal Society Open Science subject to minor revision in accordance with the referee suggestions. Please find the referees' comments at the end of this email.

The reviewers and handling editors have recommended publication, but also suggest some minor revisions to your manuscript. Therefore, I invite you to respond to the comments and revise your manuscript.

- Ethics statement

- Data accessibility

<http://datadryad.org/submit?journalID=RSOS&manu=RSOS-190337>

- **Competing interests**

- **Authors' contributions**

- **Acknowledgements**

- **Funding statement**

Because the schedule for publication is very tight, it is a condition of publication that you submit the revised version of your manuscript before 26-Jul-2019. Please note that the revision deadline will expire at 00.00am on this date. If you do not think you will be able to meet this date please let me know immediately.

on behalf of Dr Asha de Vos (Associate Editor) and Kevin Padian (Subject Editor)
 openscience@royalsociety.org

Associate Editor Comments to Author (Dr Asha de Vos):

Please consider modifying the paper as per comments from Reviewer 1. Thank you.

Reviewer comments to Author:

Reviewer: 1

Reviewer Comments: Owen-et-al_Humps_RSOS-2019

Title might be refined as it seems a bit inverted. Doesn't transmission of song occur before "cultural convergence"? How do you know it's simultaneous given the very limited period of time during which acoustic samples were collected and the sparse sampling locations over a vast ocean area? Title simplified might better read: "Migratory convergence facilitates transmission and cultural convergence of humpback whale song"?

Summary seems quite selective in terms of cultural transmission: vertebrates, humans, cetaceans. I suggest a very careful review of the wording for detailed accuracy and revise as needed.

Examples:

Line 7: Given that "birds to humans" implies only vertebrate species are considered to engage in cultural transmission behaviors, I suggest inserting "vertebrate" before "taxa".

Line 9: Consider changing "sing" to "perform" so as to read " perform complex, culturally transmitted song displays that can change in ... " The way this is presently stated, it reads as if all populations change songs in both ways, when this has not yet been demonstrated in all populations.

Line 12: Areas of migratory contact remain unknown ... "across vast oceanic regions". This is an example of unnecessary hyperbole. "these areas of contact remain unknown" - Isn't it more the case that the specific areas of acoustic contact remain uncertain? Look at tagging results that show general, broad migratory "avenues" in other "vast" ocean regions - e.g. North Pacific and North Atlantic.

Line 13: "Here, we linked song recordings from..." You actually did not "link" song recordings (Recording is a mechanism, and what does "link" mean?). Isn't it more accurate to say that you identified songs recorded from the Kermadec Island area during spring migration as having similarities to songs recorded from a handful of South Pacific island areas during the winter? You then interpreted such similarities as evidence of song transmission between populations.

Lines 15-17: Sentence would benefit from a rewrite; in its present form it's a bit convoluted and unclear. Delete "in recordings" as this is assumed. Songs don't demonstrate a mechanism, though they might imply one or are a manifestation of one. Songs are a behavior that you recorded and analyzed for similarities, and from those similarities you derived some conclusions: similarities in song themes from the Kermadec Island area and South Pacific island areas indicate/suggest/are consistent with the conclusion that

Introduction:

Various, minor grammatical edits needed; for example, commas between multiple adjectives.

Lines 43-44: Whale populations don't "pass songs" onto another populations. Isn't it more appropriate to say that individuals in one population hear songs from another population and adopt song components into their repertoire.

Lines 54-55: Based on the literature and my own experiences, I would agree that songs are most frequently produced and recorded on winter breeding grounds, but less frequently while whales are migrating.

Line 59: "pinpointing" is jargon. Why not simply use "identifying"?

Lines 70-72: I suggest merging the first and second sentences so as to read something along the lines: "Here we hypothesised that if individual males migrated into the Kermadecs from multiple wintering sites during their southward migration (September and October), we should see some evidence of the cultural processes, song transmission or convergence."

Section 3.1, Song Recordings (Lines 85-103): Section needs some further info on recording conditions (e.g. depth of hydrophone, depth of water, sound velocity profile) to help reader better understand the range of song detection as a function of frequency and quality of recordings (e.g. low-frequency stripping). Also, last sentence seems to contradict previous sentences - that there was not a common sampling rate. For spectrogram display please state time, frequency and grey-scale level resolutions. Somewhere the text would benefit from discuss and deal with the issue of range of detection for singers.

Text says that Kermadecs' acoustic data consisted of 12 recording sessions throughout 13 days, but Table S1 lists 7 days, 2-8 October 2015. Please clarify. The total amount of acoustic recordings for all sites was only 19 hours and 9.5 minutes yielding song material from an estimated 91 singer. That's an average of only 12.6 mins of song per singer, which means your data does NOT incorporate song variability. You must underscore this very low sampling of song variance in your Abstract, Summary, Discussion and Conclusion.

Section 3.2 - 3.5: Song transcription (Lines 106-119): Song transcription process includes a certain level of subjectivity, which needs to be clearly stated. Kermadec Island recordings yielded 39 "individual singers" at unit-level and 17 at theme level from 12 recordings.

What's a "phrase string" - same as a song?

The supplemental acoustic song samples are only about a minute long, so one cannot conclude very much about auditory song similarity and variability from such short samples. Table 1 needs to list the duration of the singing for each "singer #". How does one know whether the same singer contributed more than once to the data listed in Table 1?

What's a "hybrid singer"? = song composed from phrase types recorded from singers recorded at multiple wintering sites?

How do authors rectify-integrate delay in a whale's acquisition of new song material with his actual singing performance that includes that new material? Is it assumed that individuals assimilate novel songs into singing behavior immediately?

This paper is based on a very small snippet, an auditory wink-blink into this phenomenon of

song sharing, acquisition, and “acoustic polymorphism.” The authors must make this critical fact (of a very limited data set) absolutely clear from the beginning of the manuscript, especially given that they write as if they have sampled a “vast” expanse of the South Pacific, when in fact they have not. Sampling was sparse over time and space and thus potentially subject to strong levels of aliasing depending on one’s level of analysis and interpretation.

Discussion

Line 268: Why is it appropriate to declare Kermadecs a significant location? Isn’t this the first time that this type of song comparison has been done in this region? Why wouldn’t this phenomenon be occurring all over the South Pacific, or is it assumed to occur only around islands? Aren’t you recording around islands because it’s easier and logistically feasible? The paper needs to adequately place these intriguing, initial results within the broader context of the South Pacific region and not write as if these are the final chapter, when in fact this is really the preface.

Line 270: The authors often take liberty to make these types of sweeping statements about the vast distances over which samples were taken so as to make it seem as if they sampled throughout that expanse, when in fact they did not! 5-6 sample sites across a 6000 km expanse of ocean is, by definition spatially aliased. So, please pull back a bit on the over exuberance of these preliminary findings, and change your focal depth such that you interpret these insights as very early discoveries into a vastly complex phenomenon. Otherwise, what words will you have left to use when the even greater discoveries emerge as you all brush off the next layer in this geographical, acoustic expedition.

Line 281 – Yes, finally, this is an important point, but you never develop it! Why not make the obvious parallel link to oscine songbirds that will sing novel song material many months after exposure to that new material? Acoustic memory and song acquisition is not new in the universe of animal song.

Section 5.1, Lines 283-293: It is not surprising that not all wintering songs were detected in such a small sample over such a short period of time! The reliance to Riekkola et al. 2018 suffers the same problem since samples in that paper were only acquired during 12 days in 2015.

This entire puzzle is intriguing but suffers from spatial and temporal aliasing (Yes, I’m being redundant!) that needs to be explicitly recognized. This would not necessarily detract from the import of this study, while also appropriately placing its results in a much more honest and appropriate scientific context.

Section 5.3, Lines 314-338: “captured a whale in the process of learning a new song display” = really worded like this? How do you know that this whale was in the process of learning a new song when you have only a crumb of recording for this animal? You, don’t, so you cannot make such a definitive statement. The self-recognition refs (6 and 11) do not really support this statement. Again, claiming that these few observations from this location are evidence that the Kermadecs are a significant location is premature, and such self-centered awards are an unnecessary distraction for a reader. Let the reader decide if these are “significant” – whatever that means in the broader context of this paper. Instead – why does the song move east? Magnetism? Geothermal? Oceanographics?

It would be extremely valuable if the authors would come up with a better way of illustrating the Spatio-temporal dynamics of this phenomenon. The Figure 1 illustration is helpful, but static. None of the illustrations are really novel, and thus they are not an effective means by which to allow your audience to visualize, and thus readily comprehend the dynamics of the cultural song-learning, song-sharing phenomenon through time and space.

Lines 327-328: The LSI analysis did not “indicate” this. It “suggested” something that is consistent with a hypothesis at the authors might want to interpret as an indication, but using “indicate” is not supported by the very scant data or their analysis. This text is another example where the authors go beyond the boundaries and overinterpret their results. Unacceptable!

Lines 335-338: “Given that humpback [6],... culture” This text is really too much! There is no direct evidence, just pure speculation, that humpbacks have any of the standard mechanisms for human speech or oscine songbird song acquisition. Implying humpback song research as a way of exploring human language is preposterous as well as amazingly naïve; at best it implies that the authors are unfamiliar with those deep bodies of scientific discovery. This text is totally a distraction, unnecessary and a diversion from the results of this paper, and the authors have done nothing to support such a claim except reference a previous interesting paper that superficially speculates.

Note, line 332, “boundaries” is plural, so “gives” needs to be “give.”

Lines 342-350: The first sentence is another misstatement and overstatement. You do not know anything about what any of the singers you recorded at the Kermadecs were singing before or after your brief song samples, or what songs they were exposed to. You only know what they sang when you recorded them. Therefore, you do not have enough data, no matter how complex the analysis, to state that you have demonstrated that convergence and/or song learning occurred at the Kermadecs. Rather, you have shown that your results are consistent with a conclusion/hypothesis that these occurred. Stating your results correctly does not diminish their importance, and you should be satisfied with these preliminary insights. Overstating your results is neither wise or acceptable.

What’s evidence have you provided that indicates “song evolution”? How can you claim this from such a limited sample. How do you know that songs “originated” on a wintering ground from these data? This all sounds very circular! All you know is that you recorded songs that were similar to those recorded someplace else. Since the whales are able to acquire songs, you don’t know that songs recorded somewhere else originated there. If animals are exposed to a song type in area A, but wait to sing that song many months later in area B; what’s the chicken and what’s the egg.

Line 347: I argue that “many” should be “multiple”. Since you do not have an adequate time series of samples from the individual singer at the same location throughout the song learning period, you cannot state that song learning is occurring at the Kermadecs site; you can only state that singers were recorded singing songs that shared characteristics of songs recorded at other sites. You can state that these results are consistent with the hypothesis that

This last comment can be applied throughout this paper, which is full of over-interpretations from the very limited data set. It’s an important observation, but cannot claim all these claims!!

Reviewer: 2

I think this is good paper which continues on from previous findings of Garland et al. The methodology described in the paper to measure cultural transmission of Humpback Song seems to be a robust, and it would be good if it was adopted by other researchers covering other regions. The only issue I had was really minor to do with the colo(u)r selection in Figure 1. The blue/greens used for song types 1a, 1b and 2 are very close and it is hard to discriminate them in the ellipses surrounding the islands. I would also be interested in how song complexity plays a role as well as to why cultural transmission only seems to travel from West to East is it just to do with relative stock size or is there something else occurring?

Author's Response to Decision Letter for (RSOS-190337.R0)

See Appendix A.

Decision letter (RSOS-190337.R1)

30-Jul-2019

Dear Dr Garland,

I am pleased to inform you that your manuscript entitled "Migratory convergence facilitates cultural transmission of humpback whale song" is now accepted for publication in Royal Society Open Science.

on behalf of Dr Asha de Vos (Associate Editor) and Kevin Padian (Subject Editor)
openscience@royalsociety.org

Appendix A

Manuscript RSOS-190337 entitled "Migratory convergence facilitates cultural transmission of humpback whale song"

- We thank the reviewers for their detailed comments to improve and streamline our manuscript. As suggested by the Associated Editor, we have focused our improvements following comments from Reviewer 1. Specifically, we have removed a number of overstatements and softened the language used throughout the manuscript.
- All line numbers refer to the clean manuscript (changes are underlined here). Also see tracked changes manuscript.

Reviewer: 1

Reviewer Comments: Owen-etal_Humps_RSOS-2019

This is another very interesting paper appropriately exploring humpback song and singing behavior as a cultural phenomenon. It is part of a growing collection of papers on the subject, primarily based on Southern Hemisphere humpback populations. This paper provides some novel and intriguing results and is worthy of publication. My biggest issue is with the tendency of the authors to overstate their results from a very small amount of data. These overstatements are unnecessary and a distraction given that the results speak for themselves, so to speak.

- We have amended the manuscript throughout following the reviewer's suggestions to alleviate this issue (see below). For example, the word 'showed' has been changed to 'suggested' (e.g., L318), and the word 'may' has been included at appropriate locations (e.g., L57, L69). The phrase 'our results are consistent with the conclusion that' has also been incorporated (e.g., L385).

Title might be refined as it seems a bit inverted. Doesn't transmission of song occur before "cultural convergence"? How do you know it's simultaneous given the very limited period of time during which acoustic samples were collected and the sparse sampling locations over a vast ocean area? Title simplified might better read: "Migratory convergence facilitates transmission and cultural convergence of humpback whale song"?

- We have simplified the title to "Migratory convergence facilitates cultural transmission of humpback whale song."

Summary seems quite selective in terms of cultural transmission: vertebrates, humans, cetaceans. I suggest a very careful review of the wording for detailed accuracy and revise as needed.

Examples:

Line 7: Given that "birds to humans" implies only vertebrate species are considered to engage in cultural transmission behaviors, I suggest inserting "vertebrate" before "taxa".

- Amended as suggested L4. "Cultural transmission of behaviour is important in a wide variety of vertebrate taxa from birds to humans."

Line 9: Consider changing "sing" to "perform" so as to read ".... perform complex, culturally transmitted song displays that can change in ...". The way this is presently stated, it reads as if all populations change songs in both ways, when this has not yet been demonstrated in all populations.

- Changed as suggested L6-8. "Male humpback whales (*Megaptera novaeangliae*) perform complex, culturally transmitted song displays that can change both evolutionarily (through accumulations of small changes) or revolutionarily (where a population rapidly adopts a novel song)."

Line 12: Areas of migratory contact remain unknown ... “across vast oceanic regions”. This is an example of unnecessary hyperbole. “these areas of contact remain unknown” – Isn’t it more the case that the specific areas of acoustic contact remain uncertain? Look at tagging results that show general, broad migratory “avenues” in other “vast” ocean regions - e.g. North Pacific and North Atlantic.

- *Sentence amended to reflect comments L9-10. “Acoustic contact on migratory routes may provide a mechanism for cultural revolutions of song, yet these areas of contact remain uncertain.”*

Line 13: “Here, we linked song recordings from...” You actually did not “link” song recordings (Recording is a mechanism, and what does “link” mean?). Isn’t it more accurate to say that you identified songs recorded from the Kermadec Island area during spring migration as having similarities to songs recorded from a handful of South Pacific island areas during the winter? You then interpreted such similarities as evidence of song transmission between populations.

- *Reworded. L10-12. “Here, we compared songs recorded from the Kermadec Islands, a recently discovered migratory stopover, to multiple South Pacific wintering grounds.”*

Lines 15-17: Sentence would benefit from a rewrite; in its present form it’s a bit convoluted and unclear. Delete “in recordings” as this is assumed. Songs don’t demonstrate a mechanism, though they might imply one or are a manifestation of one. Songs are a behavior that you recorded and analyzed for similarities, and from those similarities you derived some conclusions: similarities in song themes from the Kermadec Island area and South Pacific island areas indicate/suggest/are consistent with the conclusion that

- *Rewritten as suggested. L12-15. “Similarities in song themes from the Kermadec Islands and multiple wintering locations (from New Caledonia across to the Cook Islands) suggest a location allowing cultural transmission of song eastward across the South Pacific, active song learning (hybrid songs), and the potential for cultural convergence after acoustic isolation at the wintering grounds.”*

Introduction:

Various, minor grammatical edits needed; for example, commas between multiple adjectives.
- *Checked and edited as needed.*

Lines 43-44: Whale populations don’t “pass songs” onto another populations. Isn’t it more appropriate to say that individuals in one population hear songs from another population and adopt song components into their repertoire.

- *The sentence has been reworded. L43-45 – note that it is entire songs that are adopted, not individual components. “It is a clear example of large-scale horizontal cultural transmission, where a population rapidly adopts a novel song introduced from a neighbouring population [11], and then the next adjacent population adopts the novel song, and so on in a population level transmission chain [9].”*

Lines 54-55: Based on the literature and my own experiences, I would agree that songs are most frequently produced and recorded on winter breeding grounds, but less frequently while whales are migrating.

- *Songs are frequently produced and recorded on migration in the South Pacific region (e.g., off the east coast of Australia). The sentence has been amended to clarify this distinction and a reference added. L55-56. “Humpback whale song is most frequently produced and recorded on the winter breeding grounds [15] and while the whales are migrating to and (particularly in the South Pacific) from their wintering grounds [11].”*

Line 59: “pinpointing” is jargon. Why not simply use “identifying”?

- *Amended as suggested. L61-63. “All these mechanisms could occur in the South Pacific*

[19], but capturing such events and/or identifying important geographic locations in Oceania (western and central South Pacific) remains challenging given the open ocean migratory range of humpback whales.”

Lines 70-72: I suggest merging the first and second sentences so as to read something along the lines: “Here we hypothesised that if individual males migrated into the Kermadecs from multiple wintering sites during their southward migration (September and October), we should see some evidence of the cultural processes, song transmission or convergence.”

- *The sentences have been combined following the reviewer’s suggestion. L75-77. “Here, we hypothesised that if males do migrate past the Kermadec Islands from multiple wintering grounds during their southward migration (September and October 2015), we should see some evidence of the cultural processes, song transmission, and/or convergence.”*

Section 3.1, Song Recordings (Lines 85-103): Section needs some further info on recording conditions (e.g. depth of hydrophone, depth of water, sound velocity profile) to help reader better understand the range of song detection as a function of frequency and quality of recordings (e.g. low-frequency stripping). Also, last sentence seems to contradict previous sentences - that there was not a common sampling rate.

- *Additional recording information and clarification has been added to Section 3.1. Detailed set up information of fixed arrays can be found in the references but has been summarised here. The last sentence has been removed and all sampling rates are clearly stated. L90-116.*

For spectrogram display please state time, frequency and grey-scale level resolutions.

- *Additional information added to Figure caption.*

Somewhere the text would benefit from discuss and deal with the issue of range of detection for singers.

- *This information has been added, e.g., L106-107. “Such an arrangement allowed accurate tracking of singing whales within a 10 km radius [22].” L365-366. “On the wintering grounds, South Pacific humpback whales are acoustically isolated (song only travels effectively for tens of kilometers at most [46]) from other wintering grounds due to their natal site fidelity.”*

Text says that Kermadecs’ acoustic data consisted of 12 recording sessions throughout 13 days, but Table S1 lists 7 days, 2-8 October 2015. Please clarify.

- *This has been fixed (table S1).*

The total amount of acoustic recordings for all sites was only 19 hours and 9.5 minutes yielding song material from an estimated 91 singer. That’s an average of only 12.6 mins of song per singer, which means your data does NOT incorporate song variability. You must underscore this very low sampling of song variance in your Abstract, Summary, Discussion and Conclusion.

- *We have now stated this explicitly in the text. L114-116. “While we endeavoured to analyse all available recordings to capture as much singer variability as possible, the available data represent a snapshot in time of a small (but broadly representative, due to strong song matching [9]) sample of singers from each location.”*

Section 3.2 – 3.5: Song transcription (Lines 106-119): Song transcription process includes a certain level of subjectivity, which needs to be clearly stated. Kermadec Island recordings yielded 39 “individual singers” at unit-level and 17 at theme level from 12 recordings.

- *We have amended the opening sentence of section 3.3 Quantifying Units, to address subjectivity. L136-138. “To reduce subjectivity and aid in a robust, repeatable unit and theme classification, the highest quality recording at each recording location was selected to measure units for subsequent inclusion in a quantitative analysis, aiming to maximise consistent naming/classification of units between locations and song types.”*

What's a "phrase string" – same as a song?

- We have clarified this term in the text. L170-173. *"The LSI was initially run here with all unit sequence data (i.e., the sequence of units making up each phrase, hereafter called a 'phrase string') from every singer and clustered to validate the qualitative assignment of each phrase string to a phrase type (and higher-level theme grouping)."*

The supplemental acoustic song samples are only about a minute long, so one cannot conclude very much about auditory song similarity and variability from such short samples. Table 1 needs to list the duration of the singing for each "singer #". How does one know whether the same singer contributed more than once to the data listed in Table 1?

- The assumptions surrounding the assignment of 'singer' in Table 1 are stated in the text L129-133. *"When multiple singers were present in a recording or a singer was silent for more than three minutes before the song continued, it was not always possible to confirm the same singer was resuming. To avoid ambiguity, the subsequent phrases were labelled with a letter (e.g., KI01S1a) and the strings of phrases were analysed separately. This resulted in 39 'individual' singers in the Kermadec dataset from the 12 initial recordings (table S1)."*

- The following was added to the caption of table 1. *"See section 3.2 Song transcription for designation of 'individual' Kermadec singers and table S2 for further information on singers."*

- We have added table S2 which includes additional information about the 'singers' listed in Table 1.

What's a "hybrid singer"? = song composed from phrase types recorded from singers recorded at multiple wintering sites?

- We have clarified this term in the text. L249-251. *"One singer from the Kermadecs sang themes from song type 2 along with a phrase variant of one theme from song type 1a (i.e., hybrid singer, defined as a singer who sings a song containing themes from two song types [6, 11])."*

How do authors rectify-integrate delay in a whale's acquisition of new song material with his actual singing performance that includes that new material? Is it assumed that individuals assimilate novel songs into singing behavior immediately?

- This is a very good point and one of the unanswered questions in our understanding of humpback whale song learning. We have therefore been conservative surrounding this point and have stated what we have found with the minimum of assumptions.

This paper is based on a very small snippet, an auditory wink-blink into this phenomenon of song sharing, acquisition, and "acoustic polymorphism." The authors must make this critical fact (of a very limited data set) absolutely clear from the beginning of the manuscript, especially given that they write as if they have sampled a "vast" expanse of the South Pacific, when in fact they have not. Sampling was sparse over time and space and thus potentially subject to strong levels of aliasing depending on one's level of analysis and interpretation.

- We thank the reviewer for this important comment. We have followed the alternative wordings where suggested by the reviewer to ensure this point is acknowledged. We have included the available recordings in analysis to ensure the largest possible sample size and have analysed data at two levels within the song hierarchy.

Discussion

Line 268: Why is it appropriate to declare Kermadecs a significant location? Isn't this the first time that this type of song comparison has been done in this region? Why wouldn't this phenomenon be occurring all over the South Pacific, or is it assumed to occur only around islands? Aren't you recording around islands because it's easier and logistically feasible? The paper needs to adequately place these intriguing, initial results within the broader context of

the South Pacific region and not write as if these are the final chapter, when in fact this is really the preface.

- *The sentence has been reworded to reflect this comment. L299-300. “This is clearly one (of potentially multiple) important locations for the cultural transmission of humpback song.”*

Line 270: The authors often take liberty to make these types of sweeping statements about the vast distances over which samples were taken so as to make it seem as if they sampled throughout that expanse, when in fact they did not! 5-6 sample sites across a 6000 km expanse of ocean is, by definition spatially aliased. So, please pull back a bit on the over exuberance of these preliminary findings, and change your focal depth such that you interpret these insights as very early discoveries into a vastly complex phenomenon. Otherwise, what words will you have left to use when the even greater discoveries emerge as you all brush off the next layer in this geographical, acoustic expedition.

- *We have reworded the sentence (but also take the reviewer’s overall point). L300-303.*

“Owing to fine-scale differences identified in the song recordings from the wintering grounds analysed here, it was possible to match the songs recorded at the Kermadecs to New Caledonia, Tonga, Niue, and the Cook Islands, a pattern mirrored by data on genetically and photographically identified individual whales in the same area [21].”

- *We have covered the major, known wintering locations where whales using this migration route aggregate in this study (where data were available). Future work/surveys will help assess whether there are any other unidentified aggregations and continue to add to this emerging picture.*

Line 281 – Yes, finally, this is an important point, but you never develop it! Why not make the obvious parallel link to oscine songbirds that will sing novel song material many months after exposure to that new material? Acoustic memory and song acquisition is not new in the universe of animal song.

- *The line numbers do not seem to line up here. I am unsure if the reviewer is referring to L309-312 – hybrid song/song learning, or L312-315 - remembering song until the next season (I suspect it is this).*

- *Hybrid/learning is explored in more detail in section 5.3. Song learning and convergence.*

- *We have added the following text to link to acoustic memory in birds. L312-315.*

“Furthermore, as humpback whales sing less frequently on the feeding grounds [19], and as far as we know, do not aggregate at the Kermadecs during the northward migration, the song learnt during the southward migration needs to be remembered until the next breeding season (akin to song memory in oscine songbirds [36]).”

Section 5.1, Lines 283-293: It is not surprising that not all wintering songs were detected in such a small sample over such a short period of time! The reliance to Riekkola et al. 2018 suffers the same problem since samples in that paper were only acquired during 12 days in 2015.

- *Section 5.1 presents one possible explanation as to why they may be absent.*

This entire puzzle is intriguing but suffers from spatial and temporal aliasing (Yes, I’m being redundant!) that needs to be explicitly recognized. This would not necessarily detract from the import of this study, while also appropriately placing its results in a much more honest and appropriate scientific context.

- *We have endeavoured to reign in some of the statements throughout the manuscript to assist in this.*

Section 5.3, Lines 314-338: “captured a whale in the process of learning a new song display” = really worded like this? How do you know that this whale was in the process of learning a new song when you have only a crumb of recording for this animal? You, don’t, so you cannot make such a definitive statement. The self-recognition refs (6 and 11) do not really support this statement. Again, claiming that these few observations from this location are evidence

that the Kermadecs are a significant location is premature, and such self-centered awards are an unnecessary distraction for a reader. Let the reader decide if these are “significant” – whatever that means in the broader context of this paper. Instead – why does the song move east? Magnetics? Geothermal? Oceanographics?

- Reworded. L352-355. *“Hybrid songs are rare and likely short lived [6, 11], so this hybrid song, which we have likely captured some part of the process by which singers change their song display from an older to a new song version, suggests that the Kermadecs are a location where song learning occurs.”*

- We have added and expanded on the concept of easterly transmission, as requested. L355-363. *“Whales stay on average 4.6 days at Raoul Island (R.C. unpublished data), which potentially allows for exposure to song for several days. Such song exposure may allow both the learning and transmission of population-specific phrase types among populations, and acquisition of new songs facilitating the easterly transmission of these songs across the South Pacific. While this addresses how songs may spread east, the question of why songs only spread in an easterly direction remains unresolved. One possible explanation is the difference in population sizes; the large eastern Australian population may influence the smaller South Pacific populations more than they influence it [9]. Future research examining population size, density and transmission, potentially using agent-based models [45], may help to answer this question.”*

It would be extremely valuable if the authors would come up with a better way of illustrating the Spatio-temporal dynamics of this phenomenon. The Figure 1 illustration is helpful, but static. None of the illustrations are really novel, and thus they are not an effective means by which to allow your audience to visualize, and thus readily comprehend the dynamics of the cultural song-learning, song-sharing phenomenon through time and space.

- We have updated the figure following comments from Reviewer 2.

Lines 327-328: The LSI analysis did not “indicate” this. It “suggested” something that is consistent with a hypothesis at the authors might want to interpret as an indication, but using “indicate” is not supported by the very scant data or their analysis. This text is another example where the authors go beyond the boundaries and overinterpret their results. Unacceptable!

- Reworded. L371-373. *“The fine-scale LSI analysis is consistent with the hypothesis that the reestablishment of acoustic contact at the Kermadecs may have initiated the merging of these population-level differences in song (type 2) to a single norm, resulting in a degree of cultural convergence.”*

Lines 335-338: “Given that humpback [6],... culture” This text is really too much! There is no direct evidence, just pure speculation, that humpbacks have any of the standard mechanisms for human speech or oscine songbird song acquisition. Implying humpback song research as a way of exploring human language is preposterous as well as amazingly naïve; at best it implies that the authors are unfamiliar with those deep bodies of scientific discovery. This text is totally a distraction, unnecessary and a diversion from the results of this paper, and the authors have done nothing to support such a claim except reference a previous interesting paper that superficially speculates.

- Sentence deleted.

Note, line 332, “boundaries” is plural, so “gives” needs to be “give.”

- Changed.

Lines 342-350: The first sentence is another misstatement and overstatement. You do not know anything about what any of the singers you recorded at the Kermadecs were singing before or after your brief song samples, or what songs they were exposed to. You only know what they sang when you recorded them. Therefore, you do not have enough data, no matter how complex the analysis, to state that you have demonstrated that convergence and/or song learning occurred at the Kermadecs. Rather, you have shown that your results are consistent with a conclusion/hypothesis that these occurred. Stating your results correctly does not

diminish their importance, and you should be satisfied with these preliminary insights. Overstating your results is neither wise or acceptable.

- Reworded. L385-386. “Here, our results are consistent with the conclusion that song learning, transmission and potentially convergence occurs at the Kermadecs, where migratory routes from multiple populations overlap.”

What's evidence have you provided that indicates “song evolution”? How can you claim this from such a limited sample. How do you know that songs “originated” on a wintering ground from these data? This all sounds very circular! All you know is that you recorded songs that were similar to those recorded someplace else. Since the whales are able to acquire songs, you don't know that songs recorded somewhere else originated there. If animals are exposed to a song type in area A, but wait to sing that song many months later in area B; what's the chicken and what's the egg.

- ‘Song evolution’ changed to ‘convergence’. L387-390. “While convergence and transmission have been shown within a population during migration and on their wintering grounds [9, 11], song exchange and convergence on a shared migratory route, and the location of such an event, remained elusive.”

Line 347: I argue that “many” should be “multiple”. Since you do not have an adequate time series of samples from the individual singer at the same location throughout the song learning period, you cannot state that song learning is occurring at the Kermadecs site; you can only state that singers were recorded singing songs that shared characteristics of songs recorded at other sites. You can state that these results are consistent with the hypothesis that

- Reworded as suggested. L392-395. “Although it is not possible to discount other potential mechanisms of song transmission [16], our results are consistent with the hypothesis of song learning on a shared migratory route, a mechanism that could drive the eastern transmission of song across the South Pacific [9].”

This last comment can be applied throughout this paper, which is full of over-interpretations from the very limited data set. It's an important observation, but cannot claim all these claims!!

Reviewer: 2

I think this is good paper which continues on from previous findings of Garland et al. The methodology described in the paper to measure cultural transmission of Humpback Song seems to be a robust, and it would be good if it was adopted by other researchers covering other regions. The only issue I had was really minor to do with the colo(u)r selection in Figure 1. The blue/greens used for song types 1a, 1b and 2 are very close and it is hard to discriminate them in the ellipses surrounding the islands.

- Thank you for your comments. We have changed the figure accordingly.

I would also be interested in how song complexity plays a role as well as to why cultural transmission only seems to travel from West to East is it just to do with relative stock size or is there something else occurring?

- Yes, this is an intriguing question for future research. We have included a few sentences discussing easterly song transmission (noted above L355-363).